# Perspectives of Primary Healthcare Workers on HIV Injectable Pre-Exposure Prophylaxis (PrEP): A Scoping Review Protocol

**DOI:** 10.3390/ijerph22060830

**Published:** 2025-05-24

**Authors:** Nomvuselelo Nomzamo Mbatha, Nomakhosi Mpofana, Dumile Gumede

**Affiliations:** 1Faculty of Health Sciences, Durban University of Technology, Durban 4001, South Africa; dumileg@dut.ac.za; 2Department of Somatology, Faculty of Health Sciences, Durban University of Technology, Durban 4001, South Africa; nomakhosim@dut.ac.za

**Keywords:** perceptions, awareness, injectable HIV pre-exposure prophylaxis (PrEp), primary healthcare workers, implementation challenges, healthcare provider perceptions, HIV prevention

## Abstract

South Africa continues to experience a high HIV prevalence, necessitating innovative prevention strategies aligned with the UNAIDS 95-95-95 targets. Long-acting injectable pre-exposure prophylaxis (PrEP), such as cabotegravir (CAB-LA), offers a promising alternative to daily oral regimens. However, the perspectives of primary healthcare workers (PHCWs)—key implementers of this intervention—remain underexplored. This scoping review aims to systematically map existing literature on PHCWs’ knowledge, awareness, perceptions, barriers, facilitators, and implementation experiences related to injectable PrEP within the South African healthcare context. The review will follow the Arksey and O’Malley framework, enhanced by Levac et al., and will be reported following PRISMA-ScR guidelines. A comprehensive search will be conducted across PubMed, Scopus, CINAHL (EBSCOhost), Web of Science, and Google Scholar, without language or date restrictions. The search strategy will employ both controlled vocabulary (e.g., MeSH and CINAHL Subject Headings) and free-text terms. Studies meeting the inclusion criteria will be managed using EndNote X20 and appraised using the Mixed Methods Appraisal Tool (MMAT) 2018 version. Data will be synthesized thematically and presented narratively and in tabular form. By consolidating PHCWs’ perspectives, this review will identify implementation challenges, training needs, and systemic barriers, informing the development of context-specific strategies for PrEP rollout. The findings are expected to support the design of effective, culturally relevant educational interventions and guide policymakers in strengthening HIV prevention efforts in high-burden settings.

## 1. Introduction

According to UNAIDS 2023, approximately 39 million [33.1 million–45.7 million] people globally were living with HIV in 2022. Sub-Saharan Africa is the most affected region, with 25.7 million people living with HIV/AIDS and 470,000 deaths. South Africa has one of the highest HIV/AIDS burdens, with 7.8 million people living with the virus and 110,000 deaths (16% of the global total). KwaZulu-Natal province in South Africa has a particularly high prevalence rate [1]. Injectable PrEP can potentially address some of the challenges associated with daily oral PrEP. It offers a more discreet and convenient option for individuals who may struggle with adherence to daily pill-taking. By providing longer-lasting protection, injectable PrEP could help bridge the gap and reach more people at risk of HIV infection [2]. It is worth noting that while injectable PrEP shows promise, it is not without its limitations [3]. It requires healthcare providers to administer the injections, which may be a barrier in some settings. Additionally, there are potential side effects and considerations for individuals who may have concerns about needles or medical procedures [4].

The development and availability of injectable PrEP (pre-exposure prophylaxis) marks a significant advancement in the fight against HIV/AIDS, particularly for populations at a higher risk of infection [3]. Injectable PrEP, administered through injections every few months, offers a promising alternative to daily oral pills, potentially improving adherence for individuals who may struggle with daily regimens [2]. This innovation can lead to higher levels of protection and help reduce new HIV infections, contributing to public health goals aimed at controlling and ultimately eradicating the virus. Moreover, the introduction of injectable PrEP is especially crucial in areas heavily affected by HIV, where traditional methods may not be as effective due to various barriers, including stigma, forgetfulness, or access to healthcare facilities [5]. This long-acting formulation could be a game changer, particularly for young people, men who have sex with men, and transgender individuals, who often face unique challenges in accessing preventive measures [6].

However, while the introduction of injectable PrEP is a promising development, it is essential to prioritize equitable access to all effective prevention methods. This involves addressing systemic barriers that disproportionately affect marginalized communities, such as socioeconomic disparities, lack of healthcare resources, and cultural stigma surrounding HIV and its prevention [7]. Public health initiatives should focus on comprehensive education, outreach programs, and policy changes that facilitate access to both injectable PrEP and other prevention strategies, such as oral PrEP, condoms, and regular testing [8]. By ensuring that these tools are available to everyone, regardless of their background or circumstances, we can move closer to achieving health equity and ultimately reducing the burden of HIV globally [9].

The rationale for reviewing primary healthcare workers’ perspectives on injectable PrEP, particularly cabotegravir (CAB-LA), stems from the growing recognition of its potential as a viable HIV prevention strategy. Existing studies indicate a preference for injectable over oral PrEP due to improved adherence and convenience, especially among populations seeking privacy and discretion [10]. The perspectives of primary healthcare workers (PHCWs) are essential because they are on the front lines of healthcare delivery and play a critical role in the successful implementation of new interventions. Their insights can help identify practical barriers and facilitators to the adoption of injectable PrEP, ensuring that the intervention is both effective and feasible in real-world settings. PHCWs’ acceptance and support are crucial for the successful rollout of injectable PrEP, as they are responsible for educating patients, administering the injections, and monitoring adherence and side effects.

In the context of South Africa, where the HIV epidemic is particularly severe, understanding the views of PHCWs is crucial [11]. South Africa has recently approved the use of long-acting injectable PrEP, such as Cabotegravir, which offers a promising new option for HIV prevention [12]. However, the implementation of this intervention in resource-limited settings presents unique challenges. These include logistical issues, such as the storage and administration of injectable medications, as well as socio-cultural barriers, such as stigma and misconceptions about HIV prevention methods [13].

Historically, reviews have focused on oral PrEP and its implementation challenges [14]. This current proposal builds on that foundation by specifically addressing injectable PrEP, which represents a significant advancement in HIV prevention strategies. The research team, comprising experts in scoping reviews and occupational health, developed the protocol. A librarian developed a highly sensitive search strategy without date or language limitations, aiming for broad access to the existing literature. All information obtained in the scoping review stages enhances transparency, thus supporting methodological replication and minimizing biases and data duplication, according to the principles of open science.

By synthesizing the perspectives of healthcare workers, this review offers valuable insights into the potential challenges and opportunities associated with the rollout of injectable PrEP. These insights will assist policymakers and healthcare providers in creating effective strategies to enhance the uptake and delivery of this promising HIV prevention tool.

Understanding healthcare workers’ insights is crucial for addressing these barriers and optimizing implementation strategies, ultimately enhancing HIV prevention efforts across diverse communities [15]. Injectable PrEP, particularly cabotegravir (CAB-LA), has emerged as a significant advancement in HIV prevention, demonstrating superior efficacy compared to oral PrEP [16]. Clinical trials have shown that consistent use of CAB-LA can reduce the risk of HIV acquisition by over 90%, making it a critical tool in combating the epidemic, especially among high-risk populations [10,17]. The World Health Organization has expanded its recommendations to include all individuals at substantial risk of HIV infection, thereby broadening the scope of PrEP beyond just key populations [18].

Despite these advancements, the implementation of injectable PrEP faces numerous challenges. Research indicates that awareness and knowledge deficits among healthcare workers (HCWs) significantly hinder the effective rollout of PrEP services. Many HCWs express concerns about patient adherence, potential behavioral risk compensation, and their readiness to deliver this new intervention [19]. Furthermore, while some studies have explored patient perspectives on injectable PrEP, there remains a lack of comprehensive understanding regarding HCWs’ attitudes, beliefs, and perceived barriers to its implementation [20]. There is a gap in the knowledge and acceptance of HIV-injectable PrEP among primary healthcare workers [21,22]. This lack of awareness and understanding may hinder the effective implementation of PrEP programs and impact the overall success of HIV prevention efforts [19].

Most existing literature has concentrated on patient experiences and preferences regarding PrEP, particularly in high-income countries. There is insufficient attention given to the perceptions of HCWs, especially in low- and middle-income countries (LMICs), where they play a crucial role as gatekeepers in the delivery of HIV prevention services [11,23]. HCWs have reported feeling inadequately trained to prescribe and manage injectable PrEP. This lack of confidence could lead to reluctance in recommending PrEP to patients, which ultimately may affect uptake [24]. This scoping review aims to map evidence available in the literature on healthcare workers’ (HCWs) perspectives regarding injectable pre-exposure prophylaxis (PrEP), specifically cabotegravir (CAB-LA), by understanding their views on acceptability, challenges, and facilitators for implementation, which may provide valuable insights to inform future training programs and policy decisions. Ultimately, enhancing HCW engagement with injectable PrEP is crucial for optimizing its integration into HIV prevention strategies globally, focusing on specific questions and objectives, and highlighting key elements such as the population, concepts, and context involved. Cabotegravir long-acting (CAB-LA) is an injectable form of pre-exposure prophylaxis (PrEP) for HIV prevention. It is administered intramuscularly, with the first two injections given four weeks apart, followed by an injection every eight weeks [25]. CAB-LA is highly effective in reducing the risk of HIV acquisition, with studies indicating a 79% relative reduction in HIV risk compared to oral PrEP [25]. CAB-LA is recommended as an additional HIV prevention option for people at substantial risk of HIV infection due to its high efficacy and the convenience of less frequent dosing compared to daily oral PrEP [10]. This makes it a valuable tool in the global effort to reduce new HIV infections and improve adherence to HIV prevention strategies [26].

### 1.1. Research Question

This scoping review aims to map evidence available in the literature on healthcare workers’ (HCWs) perspectives regarding injectable pre-exposure prophylaxis (PrEP), specifically cabotegravir (CAB-LA), by understanding their views on acceptability, challenges, and facilitators for implementation. The research question that guides this scoping review is as follows:

What evidence exists on the knowledge, awareness, perceptions, barriers, and implementation experiences of primary healthcare workers in South Africa regarding the use of injectable HIV pre-exposure prophylaxis (PrEP)?

What are healthcare workers’ perceptions regarding the acceptability of injectable PrEP among their patients?What challenges do healthcare workers face in recommending and administering injectable PrEP?What facilitators do healthcare workers identify that could enhance the implementation of injectable PrEP in clinical settings?What training and support mechanisms do healthcare workers believe are necessary for the effective delivery of injectable PrEP?

**General Objective**: To provide a comprehensive understanding of primary healthcare workers’ (PHCWs) perspectives on the use of injectable pre-exposure prophylaxis (PrEP) for HIV prevention.

### 1.2. Objectives

The objectives of this scoping protocol are as follows:

To identify existing literature on healthcare workers’ views regarding injectable PrEP.

To assess the implications of these perspectives for future research and practice.

By addressing these objectives, this scoping review aims to provide a comprehensive understanding of HCW perspectives on injectable PrEP, ultimately informing strategies for its successful implementation in diverse healthcare contexts.

### 1.3. Search Strategy

(“pre exposure prophylaxis”[MeSH Terms]) AND (Knowledge OR awareness OR “Health Knowledge, Attitudes, Practice” OR attitude OR “attitude of health personnel”[MeSH Terms] OR “implementation” OR “barriers” OR “facilitators” OR “training” OR “support mechanisms”)

(“Primary Health Care”[Mesh] OR “Health Personnel”[Mesh] OR “healthcare workers” OR “health care providers” OR “nurses” OR “clinicians”) AND (“HIV”[Mesh] OR “Human Immunodeficiency Virus”) AND (“Pre-Exposure Prophylaxis”[Mesh] OR “injectable PrEP” OR “long-acting PrEP” OR “cabotegravir” OR “CAB-LA”) AND (“Attitude of Health Personnel”[Mesh] OR “perception” OR “perspective” OR “knowledge” OR “awareness” OR “experience” OR “acceptability” OR “barriers” OR “facilitators”)

The review aims to capture a comprehensive view of the challenges and opportunities associated with implementing injectable PrEP in diverse healthcare contexts by including these databases and sources.

## 2. Methodology

This scoping review protocol was designed using the JBI manual and the theoretical framework proposed by Arksey and O’Malley. In addition, the Preferred Reporting Items for Systematic Reviews and Meta-Analyses Extension for Scoping Reviews (PRISMA-ScR) guided its development (2018) [27]. The protocol follows nine steps, as shown in Figure 1.

### 2.1. Stage One: Definition and Alignment of Objectives and Research Questions

The research questions are defined using the PCC (Population, Concept, Context) framework. For this review, the population is primary healthcare workers, the concept is their perspectives on HIV injectable PrEP, and the context is the healthcare settings in South Africa. The PCC framework enables the mapping of various information, identification of potential knowledge gaps, presentation of key concepts, broad quantification of aspects of interest, and determination of practices and evidence within a specific thematic area. This framework will be used to define the research question.

For this review, we establish P as the Primary healthcare workers, C as the Perspectives of healthcare workers on HIV Injectable PrEP, including their attitudes, beliefs, knowledge, challenges, and experiences, and C as the healthcare setting where HIV Injectable PrEP is being implemented, such as clinics, hospitals, community health centerscentres, or public health programs (Table 1).

Thus, the research questions are as follows: What are healthcare workers’ perceptions regarding the acceptability of injectable PrEP among their patients? What challenges do healthcare workers face in recommending and administering injectable PrEP? What facilitators do healthcare workers identify that could enhance the implementation of injectable PrEP in clinical settings? What training and support mechanisms do healthcare workers believe are necessary for the effective delivery of injectable PrEP?

#### 2.1.1. Review Questions

Questions that will be guiding the study are as follows:What are healthcare workers’ perceptions regarding the acceptability of injectable PrEP among their patients?What challenges do healthcare workers face in recommending and administering injectable PrEP?What facilitators do healthcare workers identify that could enhance the implementation of injectable PrEP in clinical settings?What training and support mechanisms do healthcare workers believe are necessary for the effective delivery of injectable PrEP?

Thus, the research questions are as follows:

#### 2.1.2. Search Strategy

To ensure consistency between the search strategy and the review objectives, we have revised the search equation to align with the PCC framework. The revised search equation is as follows:

(“Primary Health Care”[Mesh] OR “Health Personnel”[Mesh] OR “healthcare workers” OR “health care providers” OR “nurses” OR “clinicians”) AND (“HIV”[Mesh] OR “Human Immunodeficiency Virus”) AND (“Pre-Exposure Prophylaxis”[Mesh] OR “injectable PrEP” OR “long-acting PrEP” OR “cabotegravir” OR “CAB-LA”) AND (“Attitude of Health Personnel”[Mesh] OR “perception” OR “perspective” OR “knowledge” OR “awareness” OR “experience” OR “acceptability” OR “barriers” OR “facilitators”)

#### 2.1.3. Criteria for Selection of Search Terms

The search terms were selected based on their relevance to the PCC framework and the specific objectives of the review. The criteria for selection included the following:**Relevance to Population**: Terms related to primary healthcare workers, such as “healthcare workers”, “attitude of health personnel”, and “primary health care workers”;**Relevance to Concept**: Terms related to perspectives on injectable PrEP, including “knowledge”, “awareness”, “attitudes”, “perceptions”, “implementation”, “barriers”, “facilitators”, “training”, and “support mechanisms”;**Relevance to Context**: Terms related to the healthcare settings in South Africa, ensuring the inclusion of studies conducted in relevant geographical and healthcare contexts.

#### 2.1.4. Rationale for Database Selection

The selected databases provide a comprehensive and interdisciplinary approach to literature review. PubMed is chosen for its extensive coverage of biomedical literature, while Scopus offers broad interdisciplinary coverage. Google Scholar is included for its ability to capture gray literature and diverse sources. EBSCOhost provides access to multiple databases, including CINAHL and PsycINFO, which are valuable for nursing, allied health, and behavioral sciences literature. Medline is essential for comprehensive medical literature, and CINAHL focuses specifically on nursing and allied health. PsycINFO covers behavioral and social sciences, and Web of Science offers extensive citation data across various disciplines. Additional sources of gray literature will include conference abstracts, policy documents, and reports from relevant health organizations.

This selection ensures a robust and well-rounded literature review, capturing a wide range of relevant studies and sources.

#### 2.1.5. Presentation of Results

The results will be reported following the PRISMA-ScR guidelines (2018) [27] and summarized to capture the perspectives of primary healthcare workers on HIV injectable pre-exposure prophylaxis in KwaZulu-Natal, South Africa, across various contexts. The contexts to be included are as follows:**Geographical Contexts**: Urban versus rural settings, and different regions within South Africa;**Healthcare Settings**: Clinics, hospitals, community health centers, and public health programs;**Socio-Cultural Contexts**: Influence of local health policies, cultural attitudes toward HIV, and preventive measures.

By including these contexts, the review aims to provide a comprehensive understanding of the challenges and opportunities associated with implementing injectable PrEP in diverse healthcare environments.

### 2.2. Stage Two: Development and Alignment of Inclusion Criteria

The eligibility criteria for sources of evidence included in this scoping review on the perspectives of healthcare workers regarding injectable PrEP (cabotegravir) are outlined below. These criteria are designed to ensure a comprehensive and relevant selection of literature while maintaining rigor in the review process.

The review will consider studies published from January 2018 to September 2023. This timeframe is selected to capture recent advancements in injectable PrEP, particularly after the clinical trials and subsequent approval of cabotegravir, which started gaining attention in the early 2010s. Only studies published in English will be included. This criterion is chosen due to resource constraints and the availability of translation services. However, if relevant studies in other languages are identified, they may be considered for inclusion if translation can be arranged. Both peer-reviewed articles and gray literature (such as reports, conference abstracts, and policy documents) will be included. This approach is justified as it allows for a broader understanding of healthcare workers’ perspectives beyond traditional academic publications, capturing insights from practice-based evidence and real-world experiences. Qualitative, quantitative, and mixed-method studies will be eligible for inclusion. This criterion is important because it enables a diverse range of perspectives to be analyzed, enriching the overall findings of the review. Studies must focus on healthcare workers involved in HIV prevention efforts, including, but not limited to, doctors, nurses, pharmacists, and community health workers. This focus is critical, as these individuals play a vital role in the implementation and administration of injectable PrEP. Eligible studies must explicitly address at least one of the following concepts related to injectable PrEP: acceptability, challenges/barriers to implementation, facilitators/enablers for uptake, or training needs. This ensures the review remains focused on the core objectives related to healthcare worker perspectives.

The rationale behind these eligibility criteria is to create a robust framework for identifying relevant literature that can inform our understanding of healthcare workers’ perspectives on injectable PrEP. By including a range of study designs and types of literature, the review aims to capture a comprehensive view of the challenges and opportunities associated with implementing this critical HIV prevention strategy. Additionally, focusing on recent publications ensures that the findings reflect current practices and attitudes within the rapidly evolving field of HIV prevention.

### 2.3. Stage Three: Description of Evidence Selection

Using the following keywords: “Perceptions, Awareness, Injectable HIV Pre-Exposure Prophylaxis (PrEP), Primary Health Care Workers, and South Africa”, we will create “index terms” by combining keywords and their synonyms. We will utilize Boolean operators such as “AND” and “OR” along with truncations to formulate search strings. For example, “Perceptions AND Awareness”. The search terms have been carefully chosen to align with the PCC (Population, Concept, Context) framework, which guided the development of the scoping review protocol. These terms aim to capture individual and systemic factors influencing PrEP-related practices. Lastly, we have incorporated search terms specific to South Africa to maintain geographical relevance and ensure that the findings apply to the local healthcare landscape. The combination of these terms is designed to yield a comprehensive, focused set of results that will effectively inform the scope and direction of the review.

### 2.4. Stage Four: Evidence Searching

We conducted a pilot literature search on PubMed. Based on our inclusion criteria to demonstrate the feasibility of addressing our research question through a scoping review method. The findings from this pilot search are summarized in Table 2. The search summary table details the date of the search, the search query, the database name, and the search results

#### 2.4.1. Screening Process

All search results from the databases will be exported into a reference management tool (e.g., EndNote or Covidence) to facilitate organization and deduplication. Two independent reviewers will screen the titles and abstracts of all identified records against the pre-defined eligibility criteria. Each reviewer will assess whether the study meets inclusion criteria based on keywords related to healthcare workers and injectable PrEP. Disagreements between reviewers will be resolved through discussion, and if necessary, a third reviewer will be consulted. Studies that pass the initial screening will be retrieved in full text. The same two reviewers will independently evaluate the full texts against the eligibility criteria. Reasons for exclusion at this stage will be documented for transparency.

#### 2.4.2. Eligibility Criteria

Sources included in the review must meet the following criteria: Published between January 2018 and October 2023. Available in English. Focus on healthcare workers involved in HIV prevention efforts. Address concepts related to acceptability, barriers, or facilitators of injectable PrEP. This structured selection process aims to ensure that only relevant and high-quality studies are included in the scoping review, thereby providing a comprehensive overview of healthcare workers’ perspectives on injectable PrEP.

### 2.5. Step Five: Evidence Selection

A standardized data charting form will be created based on the key variables identified from the research questions. This form will include fields for essential information such as study characteristics (author, year, study design), participant demographics, key findings related to acceptability, barriers, facilitators, and any training needs identified by healthcare workers. The charting form will be piloted with a subset of included studies to ensure clarity and comprehensiveness before full implementation. Two independent reviewers will chart data from each included source using the standardized form. This dual-review process helps minimize errors and ensures reliability in data extraction. Each reviewer will extract data independently to maintain objectivity, and discrepancies between their extractions will be resolved through discussion or consultation with a third reviewer if necessary. The charting process is designed to be iterative; as data are extracted, additional relevant variables may be identified. The charting form can be updated accordingly to capture new insights that emerge during the review. If any uncertainties or ambiguities arise during the charting process, reviewers may reach out to the studies’ original authors for clarification or additional data. This step ensures that the extracted information is accurate and comprehensive. Once all data have been charted, a descriptive summary will be generated to synthesize findings across studies. This summary will address the review’s objectives and provide insights into healthcare workers’ perspectives on injectable PrEP.

### 2.6. Step Six: Data Extraction

The researcher will extract key information from the selected studies, such as Study characteristics (authors, year, methodology). Key findings on perceptions and awareness. Themes identified in the qualitative data (e.g., barriers to PrEP uptake, knowledge gaps). An electronic information mapping document will be used to collect applicable records for each picked article. Screeners independently complete the data table form electronically with any literature that has characteristics and key information applicable to the review question (Table 3).

### 2.7. Step Seven: Evidence Analysis

The researcher will organize the extracted data to identify themes and patterns. This will involve creating summary tables or thematic maps to illustrate perceptions and awareness levels and crafting narrative summaries to highlight significant findings and gaps in the literature. The analysis will be conducted using NVivo 12 software.

A standardized data charting form will be created based on the key variables identified from the research questions. This form will include fields for essential information such as study characteristics (author, year, study design), participant demographics, key findings related to acceptability, barriers, facilitators, and any training needs identified by healthcare workers. The charting form will be piloted with a subset of included studies to ensure clarity and comprehensiveness before full implementation. Two independent reviewers will chart data from each included source using the standardized form. This dual-review process helps minimize errors and ensures reliability in data extraction. Each reviewer will extract data independently to maintain objectivity, and discrepancies between their extractions will be resolved through discussion or consultation with a third reviewer if necessary. The charting process is designed to be iterative; as data are extracted, additional relevant variables may be identified. The charting form can be updated accordingly to capture new insights that emerge during the review. If any uncertainties or ambiguities arise during the charting process, reviewers may reach out to the studies’ original authors for clarification or additional data. This step ensures that the extracted information is accurate and comprehensive.

#### 2.7.1. Thematic Analysis Process

The extracted data will be organized to identify themes and patterns. This will involve creating summary tables or thematic maps to illustrate perceptions and awareness levels and crafting narrative summaries to highlight significant findings and gaps in the literature. The analysis will be conducted using NVivo software.

We will employ an inductive thematic analysis approach to allow themes to emerge from the data without imposing preconceived categories. The steps involved in the thematic analysis process are as follows:Familiarization with Data: Reviewers will read and re-read the data to become familiar with the content.Generating Initial Codes: Initial codes will be generated based on significant features of the data. These codes will be applied to the entire dataset.Searching for Themes: Codes will be collated into potential themes, gathering all data relevant to each potential theme.Reviewing Themes: Themes will be reviewed and refined to ensure they accurately represent the data. This may involve merging, splitting, or discarding themes.Defining and Naming Themes: Each theme will be clearly defined and named, capturing the essence of what the theme represents.Producing the Report: A detailed report will be produced, including thematic maps, narrative summaries, and illustrative quotes from the data.

#### 2.7.2. Quality Appraisal

To assess the methodological quality of the included studies, this review will employ the Mixed Methods Appraisal Tool (MMAT), version 2018. The MMAT is specifically designed for the critical appraisal of studies included in systematic and scoping reviews that incorporate qualitative, quantitative, and mixed methods research (REF). This quality appraisal method allows for the evaluation of five categories of study designs: (1) qualitative research, (2) randomized controlled trials, (3) non-randomized studies, (4) quantitative descriptive studies, and (5) mixed methods studies. Two independent reviewers will conduct the appraisal to ensure objectivity and minimize bias. Each study will be scored based on the MMAT criteria, with ratings categorized as follows: high quality (100%), above average (75%), average (50%), and low quality (25%). Discrepancies in scoring will be resolved through discussion or consultation with a third reviewer. This quality assessment process will enhance the reliability of the review findings and support informed interpretation of the included evidence.

### 2.8. Step Eight: Presentation of Results

The results will be presented following the PRISMA-ScR guidelines using a structured approach that includes descriptive statistics, thematic synthesis, and narrative reporting. Extracted data will be analyzed to identify and organize themes relevant to the research aim, which is to map existing evidence on primary healthcare workers’ views of injectable PrEP. Key outcomes such as knowledge, awareness, perceptions, barriers, and implementation strategies will be highlighted. These findings will be contextualized with the review question and broader study objectives. A narrative summary will accompany the thematic results, and the review will be submitted for publication in a peer-reviewed journal to ensure transparency and dissemination of findings.

### 2.9. Step Nine: Summary of Evidence

In this stage, we will summarize the findings of the included studies regarding the primary objective of the scoping review: to explore the perspectives of primary healthcare workers on the implementation and delivery of HIV injectable PrEP. The extracted data will be collated and categorized according to major thematic domains, such as knowledge of PrEP, attitudes toward its use, perceived barriers, training needs, and readiness for implementation. The summarized evidence will be presented using tables and figures to facilitate clarity and comparison across studies. For example, a frequency table will highlight the number of studies reporting specific themes, while bar graphs and stacked charts will illustrate the distribution of perspectives and identify knowledge gaps across different domains.

This summary will allow us to perform the following tasks:Highlight areas where perspectives are consistently reported across contexts (e.g., strong support for PrEP efficacy);Identify domains with limited or inconsistent evidence (e.g., training and capacity-building needs);Suggest priority areas for future primary research, particularly in underrepresented regions or settings;Inform policymakers and implementation planners about healthcare worker readiness and concerns regarding the injectable PrEP rollout.

By mapping the current landscape of knowledge, this stage will also guide the development of targeted interventions, training programs, and support mechanisms aimed at improving the integration of HIV injectable PrEP into primary healthcare services.

Below are examples of how we will summarise our findings: Table 4, Figure 2, Figure 3 and Figure 4

This bar chart illustrates the frequency of different themes identified across studies regarding primary healthcare workers’ perspectives on HIV injectable PrEP.

This stacked bar chart summarizes the evidence identified in the review across key domains and visually highlights knowledge gaps where fewer studies were found. The lighter bars indicate areas needing further research.

This updated bar chart illustrates the number of studies that reported each major theme regarding healthcare workers’ perspectives on injectable PrEP for HIV prevention.

## 3. Discussion

This study aims to address the high burden of HIV, aligning with the Joint United Nations Programme on HIV/AIDS (UNAIDS) 95-95-95 targets, by exploring the interplay of factors shaping healthcare workers’ knowledge and perceptions. The goal is to propose recommendations for the PrEP education program in KwaZulu-Natal, South Africa.

The scoping review on healthcare workers’ perspectives regarding injectable pre-exposure prophylaxis (PrEP) has revealed several key findings, highlighting the acceptability, barriers, and facilitators associated with this innovative HIV prevention method. The review synthesized data from various studies, leading to the identification of prominent concepts and themes that aligned with the review questions and objectives. A significant theme across studies is the overall acceptability of injectable PrEP among various populations, including people who inject drugs (PWID), men who have sex with men (MSM), and healthcare workers.

## 4. Conclusions

The proposed scoping review protocol allows for the methodological identification of studies on the perspectives of primary healthcare workers on HIV injectable pre-exposure prophylaxis in KwaZulu-Natal, South Africa. The results will be published in peer-reviewed open-access journals to disseminate.

The researcher will comprehensively analyze the practical implications stemming from the study’s findings, specifically targeting areas such as policy, practice, and future research directions. For instance, one of the key recommendations may involve developing and implementing targeted training programs for healthcare workers to enhance their understanding and knowledge of injectable PrEP (pre-exposure prophylaxis). This initiative would not only equip healthcare professionals with vital information but also empower them to effectively educate patients about the benefits and usage of this preventive measure. Additionally, the researcher may advocate for improved communication strategies within healthcare settings to ensure that information regarding PrEP is conveyed clearly and effectively, thereby fostering a more informed patient population and promoting greater uptake of this critical health intervention. Changes to this protocol will be reported in the final version, including dates and justifications.

## Figures and Tables

**Figure 1 ijerph-22-00830-f001:**
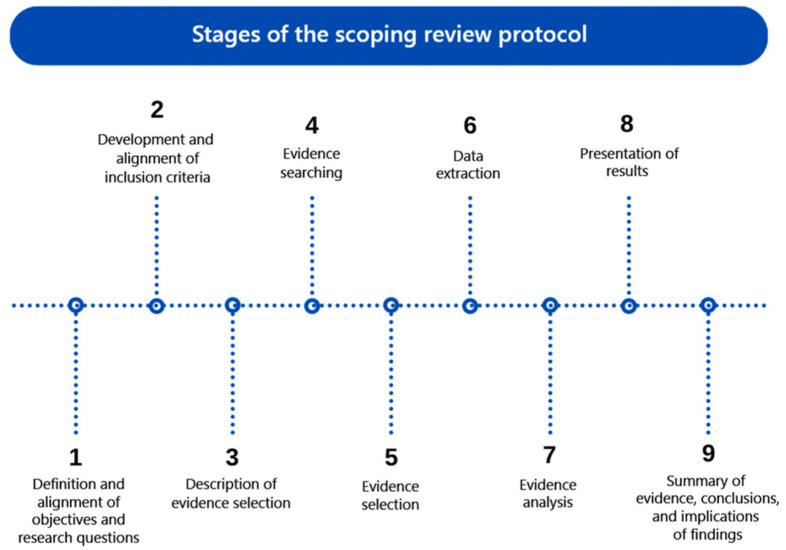
Stages of the scoping review protocol.

**Figure 2 ijerph-22-00830-f002:**
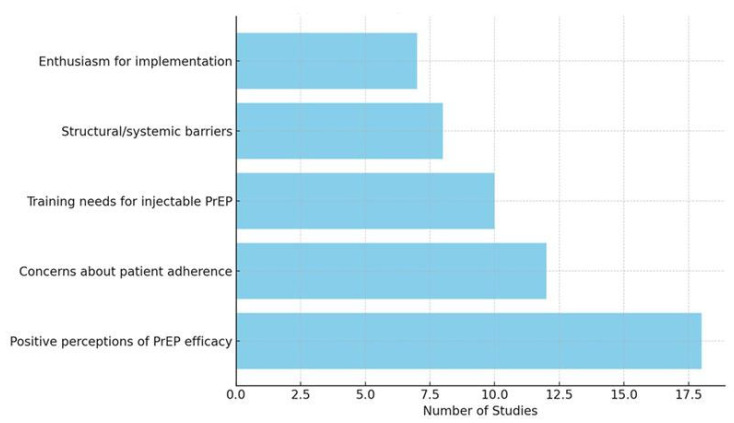
Types of perspectives identified across studies (illustrative).

**Figure 3 ijerph-22-00830-f003:**
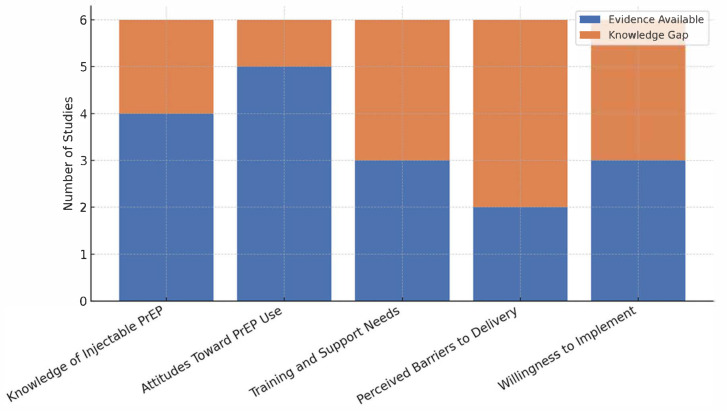
Summary of evidence and knowledge gaps.

**Figure 4 ijerph-22-00830-f004:**
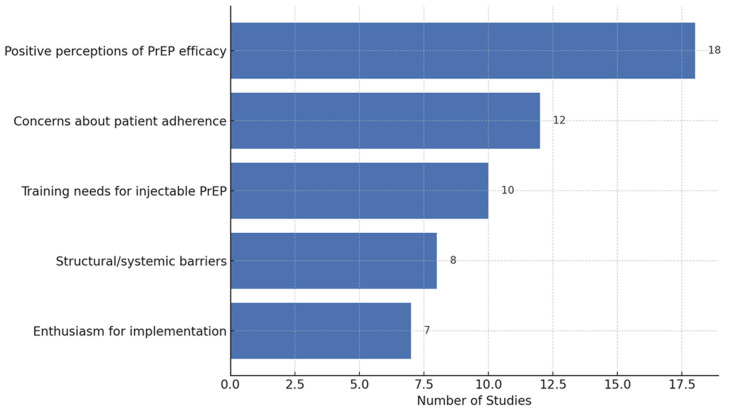
Types of perspectives identified across studies (bar chart).

**Table 1 ijerph-22-00830-t001:** Definition of concepts used in the review.

Mnemonic Elements	Concepts	Definition
**Population**	Primary healthcare workers	Healthcare workers are involved in HIV prevention, treatment, and care. This could include doctors, nurses, pharmacists, and public health professionals who interact with patients regarding HIV prevention methods. We include doctors, who diagnose and treat illnesses, prescribe medications, and provide preventive care; nurses, who provide patient care, administer medications, and support patient education and preventive measures; pharmacists, who dispense medications, provide medication counseling, and support patient adherence to prescribed treatments; community health workers, who provide basic health and medical care within the community, often focusing on preventive care and health education; and public health professionals, who work on health promotion, disease prevention, and health policy implementation at the community or population level. By explicitly defining these roles, we ensure clarity and comprehensiveness in identifying and analyzing the perspectives of primary healthcare workers on HIV injectable PrEP.
**Concept**	Perspectives of healthcare workers on HIV Injectable PrEP, including their attitudes, beliefs, knowledge, challenges, and experiences	Perspectives encompass attitudes, beliefs, knowledge, challenges, and experiences of healthcare workers concerning Injectable Pre-Exposure Prophylaxis (PrEP) for HIV prevention. This includes their knowledge about PrEP, willingness to prescribe, perceived effectiveness, barriers to implementation, and overall support for PrEP among their patients.
**Context**	The healthcare setting where HIV Injectable PrEP is being implemented, such as clinics, hospitals, community health centers, or public health programs	The setting in which healthcare workers operate which may include hospitals, clinics, community health centers, and public health programs. It might also involve considerations of geographical location, such as urban versus rural settings, and the influence of local health policies and cultural attitudes toward HIV and preventive measures.

**Table 2 ijerph-22-00830-t002:** Outcomes from PubMed search history.

Search Number	Query	Sort By	Search Details	Results	Time
4	((“primary healthcare workers” OR “healthcare personnel” OR “health workers” OR “nurses”[Mesh] OR “clinicians” OR “health personnel”[Mesh]) AND (“perspectives” OR “attitudes”[Mesh] OR “beliefs” OR “perceptions” OR “views” OR “knowledge”[Mesh] OR “awareness” OR “experience*” OR “barrier*” OR “challenge*” OR “facilitator*”) AND (“HIV”[Mesh] OR “HIV infections”[Mesh] OR “human immunodeficiency virus”) AND (“injectable PrEP” OR “long-acting PrEP” OR “cabotegravir” OR “CAB-LA” OR “Pre-Exposure Prophylaxis”[Mesh]) AND (“healthcare setting*” OR “clinics” OR “hospitals” OR “community health centre*” OR “primary health care”[Mesh] OR “public health program*”))	Most Recent	(“primary healthcare workers”[All Fields] OR “healthcare personnel”[All Fields] OR “health workers”[All Fields] OR “nurses”[MeSH Terms] OR “clinicians”[All Fields] OR “health personnel”[MeSH Terms]) AND (“perspectives”[All Fields] OR “beliefs”[All Fields] OR “perceptions”[All Fields] OR “views”[All Fields] OR “knowledge”[MeSH Terms] OR “awareness”[All Fields] OR “experience*”[All Fields] OR “barrier*”[All Fields] OR “challenge*”[All Fields] OR “facilitator*”[All Fields]) AND (“HIV”[MeSH Terms] OR “HIV infections”[MeSH Terms] OR “human immunodeficiency virus”[All Fields]) AND (“injectable PrEP”[All Fields] OR “long-acting PrEP”[All Fields] OR “cabotegravir”[All Fields] OR “CAB-LA”[All Fields] OR “Pre-Exposure Prophylaxis”[MeSH Terms]) AND (“healthcare setting*”[All Fields] OR “clinics”[All Fields] OR “hospitals”[All Fields] OR “community health centre*”[All Fields] OR “primary health care”[MeSH Terms] OR “public health program*”[All Fields])	73	09:51:59
3	((“primary healthcare workers” OR “healthcare personnel” OR “health workers” OR “nurses”[Mesh] OR “clinicians” OR “health personnel”[Mesh]) AND (“perspectives” OR “attitudes”[Mesh] OR “beliefs” OR “perceptions” OR “views” OR “knowledge”[Mesh] OR “awareness” OR “experience*” OR “barrier*” OR “challenge*” OR “facilitator*”) AND (“HIV”[Mesh] OR “HIV infections”[Mesh] OR “human immunodeficiency virus”) AND (“injectable PrEP” OR “long-acting PrEP” OR “cabotegravir” OR “CAB-LA” OR “Pre-Exposure Prophylaxis”[Mesh]) AND (“healthcare settings” OR “clinics” OR “hospitals” OR “community health centres” OR “primary health care”[Mesh] OR “public health programs”) AND (“South Africa”[Mesh] OR “South Africa”))	Most Recent	(“primary healthcare workers”[All Fields] OR “healthcare personnel”[All Fields] OR “health workers”[All Fields] OR “nurses”[MeSH Terms] OR “clinicians”[All Fields] OR “health personnel”[MeSH Terms]) AND (“perspectives”[All Fields] OR “beliefs”[All Fields] OR “perceptions”[All Fields] OR “views”[All Fields] OR “knowledge”[MeSH Terms] OR “awareness”[All Fields] OR “experience*”[All Fields] OR “barrier*”[All Fields] OR “challenge*”[All Fields] OR “facilitator*”[All Fields]) AND (“HIV”[MeSH Terms] OR “HIV infections”[MeSH Terms] OR “human immunodeficiency virus”[All Fields]) AND (“injectable PrEP”[All Fields] OR “long-acting PrEP”[All Fields] OR “cabotegravir”[All Fields] OR “CAB-LA”[All Fields] OR “Pre-Exposure Prophylaxis”[MeSH Terms]) AND (“healthcare settings”[All Fields] OR “clinics”[All Fields] OR “hospitals”[All Fields] OR “community health centres”[All Fields] OR “primary health care”[MeSH Terms] OR “public health programs”[All Fields]) AND (“South Africa”[MeSH Terms] OR “South Africa”[All Fields])	6	09:50:10
2	((“primary care” OR “primary healthcare” OR “primary health care” OR “community health services”[Mesh]) AND (“healthcare workers” OR “health workers” OR “health personnel”[Mesh] OR “nurses”[Mesh] OR “clinicians”)) AND ((“HIV”[Mesh] OR “HIV infections”[Mesh] OR “human immunodeficiency virus”) AND (“Pre-Exposure Prophylaxis” OR “injectable PrEP” OR “long-acting PrEP” OR “cabotegravir” OR “CAB-LA”)) AND ((“perceptions” OR “perspectives” OR “attitudes” OR “views” OR “knowledge” OR “awareness” OR “implementation experience” OR “barriers” OR “facilitators”))	Most Recent	(“primary care”[All Fields] OR “primary healthcare”[All Fields] OR “primary health care”[All Fields] OR “community health services”[MeSH Terms]) AND (“healthcare workers”[All Fields] OR “health workers”[All Fields] OR “health personnel”[MeSH Terms] OR “nurses”[MeSH Terms] OR “clinicians”[All Fields]) AND ((“HIV”[MeSH Terms] OR “HIV infections”[MeSH Terms] OR “human immunodeficiency virus”[All Fields]) AND (“Pre-Exposure Prophylaxis”[All Fields] OR “injectable PrEP”[All Fields] OR “long-acting PrEP”[All Fields] OR “cabotegravir”[All Fields] OR “CAB-LA”[All Fields])) AND (“perceptions”[All Fields] OR “perspectives”[All Fields] OR “attitudes”[All Fields] OR “views”[All Fields] OR “knowledge”[All Fields] OR “awareness”[All Fields] OR “implementation experience”[All Fields] OR “barriers”[All Fields] OR “facilitators”[All Fields])	92	09:43:17
1	(“Primary Health Care”[Mesh] OR “Health Personnel”[Mesh] OR “healthcare workers” OR “health care providers” OR “nurses” OR “clinicians”) AND (“HIV”[Mesh] OR “Human Immunodeficiency Virus”) AND (“Pre-Exposure Prophylaxis”[Mesh] OR “injectable PrEP” OR “long-acting PrEP” OR “cabotegravir” OR “CAB-LA”) AND (“Attitude of Health Personnel”[Mesh] OR “perception” OR “perspective” OR “knowledge” OR “awareness” OR “experience” OR “acceptability” OR “barriers” OR “facilitators”)	Most Recent	(“Primary Health Care”[MeSH Terms] OR “Health Personnel”[MeSH Terms] OR “healthcare workers”[All Fields] OR “health care providers”[All Fields] OR “nurses”[All Fields] OR “clinicians”[All Fields]) AND (“HIV”[MeSH Terms] OR “Human Immunodeficiency Virus”[All Fields]) AND (“Pre-Exposure Prophylaxis”[MeSH Terms] OR “injectable PrEP”[All Fields] OR “long-acting PrEP”[All Fields] OR “cabotegravir”[All Fields] OR “CAB-LA”[All Fields]) AND (“Attitude of Health Personnel”[MeSH Terms] OR “perception”[All Fields] OR “perspective”[All Fields] OR “knowledge”[All Fields] OR “awareness”[All Fields] OR “experience”[All Fields] OR “acceptability”[All Fields] OR “barriers”[All Fields] OR “facilitators”[All Fields])	62	09:09:50

Truncation symbol (*) used to retrieve all variants of a word stem in database searches.

**Table 3 ijerph-22-00830-t003:** Data charting form.

Title of the study
Author and year of publication
keywords
Study location
Study sector/setting
Study Aim
Study design
Study population(Primary healthcare managers, HIV nurses, HIV Counselling and Testing counsellors, and health promoters above 18)
Findings
Conclusion

**Table 4 ijerph-22-00830-t004:** Summary of included studies (Example).

Study ID	Author(s), Year	Country	Study Design	Healthcare Workers	Setting	Key Findings Related to Perspective
001	Smith et al., 2021	Kenya	Qualitative	Nurses, CHWs	Rural Clinic	Positive attitudes but limited training
002	Zhang et al., 2022	USA	Mixed methods	Physicians, Nurses	Urban Hospital	Concern about long-term adherence
003	Nguyen et al., 2023	Vietnam	Quantitative	Nurses	Community Health Center	Supportive but unclear guidelines

## Data Availability

The original contributions presented in this study are included in the article. Further inquiries can be directed to the corresponding author(s).

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
