# Peer review of "Perspectives of Primary Healthcare Workers on HIV Injectable Pre-Exposure Prophylaxis (PrEP): A Scoping Review Protocol"

_ijerph, 2025, doi:10.3390/ijerph22060830_

Round 1

Reviewer 1 Report

Comments and Suggestions for Authors

The review discusses the existing literature of the views of primary healthcare workers regarding the use of injectable PrEP for HIV prevention. 

The introduction is detailed and provides background on the purpose of the review. 

The methods is descriptive of all the stages. However, the outcomes table doesn't clearly mention criteria used for selection of the search terms. The presentation of results is not clear regarding the various contexts that would be included.

The discussion is not numbered in alignment with the text. The discussion and conclusion can be improved with references to to historical review and the proposal with the current proposal.

The references have not been cited in the text of this review.  

Author Response

Comment 1: Open Review (x) I would not like to sign my review report
( ) I would like to sign my review report Quality of English Language ( ) The English could be improved to more clearly express the research.
(x) The English is fine and does not require any improvement.  

  Yes Can be improved Must be improved Not applicable
Does the introduction provide sufficient background and include all relevant references? (x) ( ) ( ) ( )
Is the research design appropriate? (x) ( ) ( ) ( )
Are the methods adequately described? (x) ( ) ( ) ( )
Are the results clearly presented? ( ) (x) ( ) ( )
Are the conclusions supported by the results? ( ) (x) ( ) ( )

    Comments and Suggestions for Authors

The review discusses the existing literature of the views of primary healthcare workers regarding the use of injectable PrEP for HIV prevention. 

The introduction is detailed and provides background on the purpose of the review. 

The methods is descriptive of all the stages. However, the outcomes table doesn't clearly mention criteria used for selection of the search terms. The presentation of results is not clear regarding the various contexts that would be included.

The discussion is not numbered in alignment with the text. The discussion and conclusion can be improved with references to to historical review and the proposal with the current proposal.

The references have not been cited in the text of this review.  

Response 1: Please find attached document below

Reviewer 2 Report

Comments and Suggestions for Authors

Overall Assessment

The manuscript addresses an important and timely topic regarding the perspectives of primary healthcare workers (PHCWs) on HIV Injectable PrEP. The scoping review protocol is generally well-structured and follows the Arksey and O'Malley framework. However, there are several areas where improvements can be made to enhance the clarity, rigor, and overall quality of the manuscript.

Major Comments

  1. Justification and Rationale:
    • The manuscript lacks a strong justification for why perspectives of PHCWs are essential. While it mentions the importance of injectable PrEP, further elaboration on why this scoping review is particularly needed in the context of South Africa would strengthen the rationale.
    • Consider expanding on the unique challenges faced by PHCWs in implementing injectable PrEP in resource-limited settings.
  2. Methodological Rigor:
    • The manuscript claims to follow the Arksey and O'Malley framework, but it does not clearly outline how each step will be applied. A more detailed explanation of how the framework will be operationalized is needed.
    • The search strategy, while explained to some extent, lacks specificity. Include a more robust list of search terms, databases, and additional sources of grey literature. Providing a justification for each chosen database would also be beneficial.
  3. Conceptual Clarity:
    • The concepts and definitions in the PCC framework (Population, Concept, Context) need more clarity. For example, the term "primary healthcare workers" should be explicitly defined with a clear list of roles included.
    • Further, the term "perspectives" is vague. Are the authors focusing on knowledge, attitudes, beliefs, or experiences? Clearly define the sub-concepts to ensure a structured analysis.
  4. Data Extraction and Analysis:
    • While a data charting form is mentioned, there is insufficient detail on how data will be synthesized and reported. Provide a clearer description of the thematic analysis process and how the findings will be categorized.
    • Consider including a plan for assessing the quality of the included studies using a standardized tool. This would enhance the transparency and robustness of the review.

Minor Comments

  1. Abstract:
    • The abstract is informative but could be more concise. Consider reducing redundancy and improving clarity in the objective statement.
    • Keywords could be expanded to include relevant terms such as "implementation challenges," "healthcare provider perceptions," and "HIV prevention."
  1. Figures and Tables:
    • The PCC table is helpful, but adding a sample search string in the appendix would be beneficial.
Comments on the Quality of English Language

 The English could be improved to more clearly express the research.

Author Response

Please find the response attached below.

Comment 1: Open Review (x) I would not like to sign my review report
( ) I would like to sign my review report Quality of the English Language (x) The English could be improved to more clearly express the research.
( ) The English is fine and does not require any improvement.

  Yes Can be improved Must be improved Not applicable
Does the introduction provide sufficient background and include all relevant references? ( ) ( ) (x) ( )
Is the research design appropriate? ( ) (x) ( ) ( )
Are the methods adequately described? ( ) ( ) (x) ( )
Are the results clearly presented? ( ) (x) ( ) ( )
Are the conclusions supported by the results? ( ) ( ) (x) ( )

    Comments and Suggestions for Authors

Overall Assessment

The manuscript addresses an important and timely topic regarding the perspectives of primary healthcare workers (PHCWs) on HIV Injectable PrEP. The scoping review protocol is generally well-structured and follows the Arksey and O'Malley framework. However, there are several areas where improvements can be made to enhance the clarity, rigor, and overall quality of the manuscript.

Major Comments

  1. Justification and Rationale:
    • The manuscript lacks a strong justification for why perspectives of PHCWs are essential. While it mentions the importance of injectable PrEP, further elaboration on why this scoping review is particularly needed in the context of South Africa would strengthen the rationale.
    • Consider expanding on the unique challenges faced by PHCWs in implementing injectable PrEP in resource-limited settings.
  2. Methodological Rigor:
    • The manuscript claims to follow the Arksey and O'Malley framework, but it does not clearly outline how each step will be applied. A more detailed explanation of how the framework will be operationalized is needed.
    • The search strategy, while explained to some extent, lacks specificity. Include a more robust list of search terms, databases, and additional sources of grey literature. Providing a justification for each chosen database would also be beneficial.
  3. Conceptual Clarity:
    • The concepts and definitions in the PCC framework (Population, Concept, Context) need more clarity. For example, the term "primary healthcare workers" should be explicitly defined with a clear list of roles included.
    • Further, the term "perspectives" is vague. Are the authors focusing on knowledge, attitudes, beliefs, or experiences? Clearly define the sub-concepts to ensure a structured analysis.
  4. Data Extraction and Analysis:
    • While a data charting form is mentioned, there is insufficient detail on how data will be synthesized and reported. Provide a clearer description of the thematic analysis process and how the findings will be categorized.
    • Consider including a plan for assessing the quality of the included studies using a standardized tool. This would enhance the transparency and robustness of the review.

Minor Comments

  1. Abstract:
    • The abstract is informative but could be more concise. Consider reducing redundancy and improving clarity in the objective statement.
    • Keywords could be expanded to include relevant terms such as "implementation challenges," "healthcare provider perceptions," and "HIV prevention."
  1. Figures and Tables:
    • The PCC table is helpful, but adding a sample search string in the appendix would be beneficial.

Comments on the Quality of English Language

 The English could be improved to more clearly express the research.

Reviewer 3 Report

Comments and Suggestions for Authors

This paper is a research protocol of a literature review of knowledge, attitudes, and practices of HIV pre-exposure prophylaxis among health care workers. This field is relatively new, and much remains to be explored. Therefore, this study also follows the PRISMA statement and may provide effective recommendations on how health care workers should be involved in HIV pre-exposure prophylaxis in the future. I did not find major problems in the protocol, but would like to make a few suggestions.

The first is how to select keywords for literature extraction. Due to the relatively new field, it is expected that not so many useful articles will be extracted. In fact, a simplified search by using “pre-exposure prophylaxis”, “South Africa”, “HIV/AIDS”, and “Healthcare workers” yielded only six articles. Excluding “South Africa,” the number of papers increased by 19. Therefore, I believe you need to be careful when setting keywords.

Second, in the Search Details in Table 2, adding * after the keywords like “knowledge*” and “attitude*” will simplify your search formula, and may find more articles related to your study.

Third, it would be informative to present virtual tables or graphs showing the expected results in 2.9 Step Nine. This would make it easier for readers to understand your study.

As a minor comment, please check the journal's citation guidelines. [Number] and “Name, Year” are mixed throughout the manuscript.

Author Response

please see the attached response below. Open Review (x) I would not like to sign my review report
( ) I would like to sign my review report Quality of English Language ( ) The English could be improved to more clearly express the research.
(x) The English is fine and does not require any improvement.            

  Yes Can be improved Must be improved Not applicable
Does the introduction provide sufficient background and include all relevant references? (x) ( ) ( ) ( )
Is the research design appropriate? ( ) (x) ( ) ( )
Are the methods adequately described? ( ) (x) ( ) ( )
Are the results clearly presented? ( ) ( ) ( ) (x)
Are the conclusions supported by the results? ( ) ( ) ( ) (x)

Comments and Suggestions for Authors

This paper is a research protocol of a literature review of knowledge, attitudes, and practices of HIV pre-exposure prophylaxis among health care workers. This field is relatively new, and much remains to be explored. Therefore, this study also follows the PRISMA statement and may provide effective recommendations on how health care workers should be involved in HIV pre-exposure prophylaxis in the future. I did not find major problems in the protocol, but would like to make a few suggestions.

The first is how to select keywords for literature extraction. Due to the relatively new field, it is expected that not so many useful articles will be extracted. In fact, a simplified search by using “pre-exposure prophylaxis”, “South Africa”, “HIV/AIDS”, and “Healthcare workers” yielded only six articles. Excluding “South Africa,” the number of papers increased by 19. Therefore, I believe you need to be careful when setting keywords.

Second, in the Search Details in Table 2, adding * after the keywords like “knowledge*” and “attitude*” will simplify your search formula, and may find more articles related to your study.

Third, it would be informative to present virtual tables or graphs showing the expected results in 2.9 Step Nine. This would make it easier for readers to understand your study.

As a minor comment, please check the journal's citation guidelines. [Number] and “Name, Year” are mixed throughout the manuscript.

Reviewer 4 Report

Comments and Suggestions for Authors

First of all, we would like to congratulate you on the choice of topic, which is highly relevant and timely in the field of public health and HIV prevention. Below, we offer a series of recommendations aimed at improving the clarity, coherence, and methodological rigor of your protocol.
Comments on the protocol:
Abstract
The abstract of your work serves as your letter of introduction; therefore, it should highlight the most important aspects of the protocol to showcase its value. Please specify that this is a scoping review protocol and include the specific databases to be used, the thesauri employed, as well as the complete search equation. Additionally, make sure that the objective stated in the abstract exactly matches the one included in the objectives section of the project.
Introduction
The introduction is well-founded; however, I suggest that it should end with the explicit formulation of the research question. To achieve this, it would be helpful to reorganize the ideas in the final paragraph so that they naturally lead to the research question.
Objectives
I recommend clearly distinguishing between the general objective and the specific objectives. Furthermore, if the proposed search equation is:
("pre exposure prophylaxis"[MeSH Terms]) AND (Knowledge OR awareness OR "Health Knowledge, Attitudes, Practice" OR attitude OR "attitude of health personnel"[MeSH Terms]),
then the objectives should be aligned with what can realistically be achieved using that search strategy. Otherwise, it would be necessary to expand the equation by including additional concepts related to implementation or to barriers and facilitators, as the current terms alone will not be sufficient to identify relevant and adequate studies for suggesting implementation strategies.
Methodology
In the section "Definition and Alignment of Objectives and Research Questions", the population, concept, and context are correctly defined. However, the expanded search equation mentioned later (Table 2: Outcomes from PubMed Search History) includes terms that do not align with the PCC framework. For instance, while the protocol focuses on healthcare workers, terms such as "family members" or even "mindfulness" are included, which do not appear to be relevant to the research objective. We recommend revising this section carefully to ensure consistency between the search strategy and the review objectives.
In "Stage Two: Development and Alignment of Inclusion Criteria", it is stated that studies published up to October 2024 will be included, although we are currently in April 2024. This is not methodologically appropriate in a protocol unless it is clearly stated that this is a future or a living review. I recommend correcting the date or clarifying that the review will be conducted at a specific future time.
In "Stage Three: Description of Evidence Selection", it is important to justify the choice of databases used. What was the selection criterion? In addition, I suggest considering the inclusion of databases such as CINAHL, PsycINFO, or Web of Science, as they may yield more relevant studies, especially those related to healthcare workers’ perceptions, beyond what Medline alone can offer.
In "Stage Four: Evidence Searching", the following phrase appears: “We conducted a pilot literature search across various databases (like PubMed, Scopus, and Google Scholar)”. Phrases such as "like" or "for example" should be avoided. Please specify exactly which databases were consulted during the pilot search.
In the inclusion criteria, I also recommend modifying the cut-off date for study inclusion, as mentioned above.
In "Step Seven: Evidence Analysis", the software “INVIVO” is mentioned, which presumably refers to NVivo. However, the qualitative analysis approach is not described. Will it be inductive or deductive? Will any theoretical frameworks be used, such as CFIR (Consolidated Framework for Implementation Research) or TDF (Theoretical Domains Framework)? It would be advisable to clearly define the thematic analysis approach and justify it accordingly.
Lastly, in the section "Critical Appraisal Findings", a critical appraisal of the included studies is carried out. However, according to JBI recommendations, a scoping review does not necessarily require methodological quality assessment. If you decide to retain this section, it is essential to explain which tool will be used for the appraisal, how it will be applied, and how it will affect the interpretation of the results. Otherwise, I recommend removing it. 

These revisions are substantial and would significantly improve the quality of the protocol.

Best regards

Author Response

Response: please find the response attached below. 

Open Review ( ) I would not like to sign my review report
(x) I would like to sign my review report Quality of English Language ( ) The English could be improved to more clearly express the research.
(x) The English is fine and does not require any improvement.  

  Yes Can be improved Must be improved Not applicable
Does the introduction provide sufficient background and include all relevant references? ( ) ( ) (x) ( )
Is the research design appropriate? ( ) ( ) (x) ( )
Are the methods adequately described? ( ) ( ) (x) ( )
Are the results clearly presented? ( ) ( ) ( ) (x)
Are the conclusions supported by the results? ( ) ( ) ( ) (x)

    Comments and Suggestions for Authors

First of all, we would like to congratulate you on the choice of topic, which is highly relevant and timely in the field of public health and HIV prevention. Below, we offer a series of recommendations aimed at improving the clarity, coherence, and methodological rigor of your protocol.
Comments on the protocol:
Abstract
The abstract of your work serves as your letter of introduction; therefore, it should highlight the most important aspects of the protocol to showcase its value. Please specify that this is a scoping review protocol and include the specific databases to be used, the thesauri employed, as well as the complete search equation. Additionally, make sure that the objective stated in the abstract exactly matches the one included in the objectives section of the project.
Introduction
The introduction is well-founded; however, I suggest that it should end with the explicit formulation of the research question. To achieve this, it would be helpful to reorganize the ideas in the final paragraph so that they naturally lead to the research question.
Objectives
I recommend clearly distinguishing between the general objective and the specific objectives. Furthermore, if the proposed search equation is:
("pre exposure prophylaxis"[MeSH Terms]) AND (Knowledge OR awareness OR "Health Knowledge, Attitudes, Practice" OR attitude OR "attitude of health personnel"[MeSH Terms]),
then the objectives should be aligned with what can realistically be achieved using that search strategy. Otherwise, it would be necessary to expand the equation by including additional concepts related to implementation or to barriers and facilitators, as the current terms alone will not be sufficient to identify relevant and adequate studies for suggesting implementation strategies.
Methodology
In the section "Definition and Alignment of Objectives and Research Questions", the population, concept, and context are correctly defined. However, the expanded search equation mentioned later (Table 2: Outcomes from PubMed Search History) includes terms that do not align with the PCC framework. For instance, while the protocol focuses on healthcare workers, terms such as "family members" or even "mindfulness" are included, which do not appear to be relevant to the research objective. We recommend revising this section carefully to ensure consistency between the search strategy and the review objectives.
In "Stage Two: Development and Alignment of Inclusion Criteria", it is stated that studies published up to October 2024 will be included, although we are currently in April 2024. This is not methodologically appropriate in a protocol unless it is clearly stated that this is a future or a living review. I recommend correcting the date or clarifying that the review will be conducted at a specific future time.
In "Stage Three: Description of Evidence Selection", it is important to justify the choice of databases used. What was the selection criterion? In addition, I suggest considering the inclusion of databases such as CINAHL, PsycINFO, or Web of Science, as they may yield more relevant studies, especially those related to healthcare workers’ perceptions, beyond what Medline alone can offer.
In "Stage Four: Evidence Searching", the following phrase appears: “We conducted a pilot literature search across various databases (like PubMed, Scopus, and Google Scholar)”. Phrases such as "like" or "for example" should be avoided. Please specify exactly which databases were consulted during the pilot search.
In the inclusion criteria, I also recommend modifying the cut-off date for study inclusion, as mentioned above.
In "Step Seven: Evidence Analysis", the software “INVIVO” is mentioned, which presumably refers to NVivo. However, the qualitative analysis approach is not described. Will it be inductive or deductive? Will any theoretical frameworks be used, such as CFIR (Consolidated Framework for Implementation Research) or TDF (Theoretical Domains Framework)? It would be advisable to clearly define the thematic analysis approach and justify it accordingly.
Lastly, in the section "Critical Appraisal Findings", a critical appraisal of the included studies is carried out. However, according to JBI recommendations, a scoping review does not necessarily require methodological quality assessment. If you decide to retain this section, it is essential to explain which tool will be used for the appraisal, how it will be applied, and how it will affect the interpretation of the results. Otherwise, I recommend removing it. 

These revisions are substantial and would significantly improve the quality of the protocol.

Best regards

Round 2

Reviewer 2 Report

Comments and Suggestions for Authors

None

Reviewer 4 Report

Comments and Suggestions for Authors

I would like to congratulate the authors, as I believe that after applying the corrections suggested by the different reviewers, the protocol has greatly improved. I wish them excellent results and encourage them to carry out original research, both descriptive and interventional, on this important topic, given the persistent stigma that still exists.

Sincerely.